# Language Generation in the Limit

**Jon Kleinberg**
Departments of Computer Science
and Information Sciene
Cornell University
Ithaca NY

**Sendhil Mullainathan**
Booth School of Business
University of Chicago
Chicago IL

## Abstract

Although current large language models are complex, the most basic specifications of the underlying language generation problem itself are simple to state: given a finite set of training samples from an unknown language, produce valid new strings from the language that don't already appear in the training data. Here we ask what we can conclude about language generation using only this specification, without further assumptions. In particular, suppose that an adversary enumerates the strings of an unknown target language $L$ that is known only to come from one of a possibly infinite list of candidates. A computational agent is trying to learn to generate from this language; we say that the agent *generates from $L$ in the limit* if after some finite point in the enumeration of $L$, the agent is able to produce new elements that come exclusively from $L$ and that have not yet been presented by the adversary. Our main result is that there is an agent that is able to generate in the limit for every countable list of candidate languages. This contrasts dramatically with negative results due to Gold and Angluin in a well-studied model of language learning where the goal is to identify an unknown language from samples; the difference between these results suggests that identifying a language is a fundamentally different problem than generating from it.

## 1 Introduction

The recent advances in large language models (LLMs) have been remarkable, sparking active lines of theoretical work into their performance. These investigations implicitly revolve around two fundamental questions: how do we formally reason about the effectiveness of LLMs; and within such a framework, what are the core ideas at a mathematical level that enable their performance?

Answers to these questions must begin by formalizing the specification for what a generative algorithm for language should be doing. Here, we propose starting from a very basic, assumption-free, statement for such a specification: there is an unknown target language $L$, over time the algorithm sees a sequence of strings from $L$, and eventually we would like the algorithm to generate new strings from $L$ that it has not seen before.[1]

Viewed this way, it is also clear why it seems so remarkable for LLMs to be doing well at such a problem. The fully general statement of the problem feels unsolvable: if we know nothing about the unknown target language $L$, then how can a generative algorithm reliably produce valid strings from $L$ that it hasn't seen before?

---

[1] We will formalize these concepts more precisely below, but for now we can think of a language as simply any set of strings over a fixed alphabet; for example, the strings of the language could be the set of all grammatical sentences (or all well-formed expressions) according to a given grammar.

38th Conference on Neural Information Processing Systems (NeurIPS 2024).

**Language Learning in the Limit.**   In fact, there is a well-established formalism that allows us to phrase this point precisely: the classical model of language learning in the limit, formulated by Mark Gold in 1967 and fully characterized by Dana Angluin in 1980 [6, 2]. In this model, there is an unknown language $K$ that is known only to be produced by one of a list of candidate representations $R_1, R_2, R_3, \ldots$, where $L_i$ is the language produced by representation $R_i$. We can think of this list of representations as the set of all possible context-free grammars, or the set of all possible finite automata, or the set of all Turing machines with a fixed space bound, or any other generative model that produces strings; in fact, the formal result is much more general than this, in that it is sufficient to suppose that the unknown language $K$ simply comes from a countable list of candidate languages $L_1, L_2, L_3, \ldots$, and we can dispense with explicit representations altogether.[2]

In the Gold-Angluin model, an adversary enumerates the strings of $K$ one by one, and the algorithm is required after each new string to guess a language $L_i$ from the list such that $L_i = K$. If there is some finite step $t$ after which the algorithm's guess is always correct, then we say the algorithm has *identified $K$ in the limit*. Gold proved that this is impossible in general, even for simple language families such as the regular languages (i.e. those produced by finite automata), and Angluin characterized precisely those families for which it is possible, further establishing how limited they are [1, 2]. Note, crucially, that in the Gold-Angluin model, the adversary enumerates strings in $K$, but does not provide examples of strings that do not belong to $K$, nor does it allow the algorithm to ask questions about a string's membership in $K$; their point with this formalism was to focus on cases where an agent tries inferring a language purely from seeing a sufficient number of examples of strings that belong to the language. (In Section A.1, we provide a self-contained proof of the negative result for identification in the limit; while not strictly necessary for our results, we find it provides useful background and context for the problem.)

**Our Results: Language Generation in the Limit.**   These negative results of Gold and Angluin feel intuitive — how should we be able to identify a language from a finite sample when we are allowed to make essentially no assumptions about its structure? Because of this intuition, both via the Gold-Angluin model and for more informal reasons as well, the focus in language generation has naturally turned to distributional assumptions; one posits that large language models are so effective because they are able to exploit distributional probabilities of language, and from a finite set of samples they are able to estimate conditional probabilities of strings with increasing accuracy. In this way, the question moves from adversaries to probability distributions, and one seeks explanations for the effectiveness of LLMs through underlying probabilistic models.

In this paper, we offer a sharply different view: we show that in the Gold-Angluin model of adversarially produced examples, *language generation is always possible.* We will provide full details on the result and its proof beginning in the next section, but the key point at a high level is that even in an adversarial model with an unknown language $K$, language generation is a fundamentally different task than language identification: where identification asks an algorithm to eventually name a language $L_i = K$ after seeing a large enough finite sample $S$ from $K$, generation instead asks an algorithm to eventually output strings in $K - S$ after seeing a large enough $S$ from $K$. Our main result is that this difference in specifications leads to dramatic differences in what is possible in the limit; whereas the Gold-Angluin results establish that identification in the limit is impossible except in highly constrained cases, we show that generation in the limit is possible for *every* countable list of candidate languages.

**General Connections to Language Modeling.**   Our approach emphasizing theoretical properties of language generation and worst-case guarantees, in the style of the Gold-Angluin model, is a source of limitations but also a source of generality; it is therefore important to discuss how we draw stylized insights about language modeling from our theoretical formalism. Clearly, methods to design large language models in practice make extensive use of the empirical distributional properties of language, as they should. Our results don't question this design methodology; when there are genuine empirical regularities in the training data, there is no reason not to use them. Rather, our results argue that if we are looking for the essential reasons why language generation is tractable, we do not fundamentally require any empirical regularities, or indeed any probabilistic assumptions at all; there is instead a formal sense in which language generation — unlike broader learning tasks such as language

---

[2]In this more general view, we will assume that the family of languages is presented simply via a black box that for a string $w$ and an index $i$ can answer the question, "Is $w \in L_i$?"

identification — is possible even against an adversary presenting positive training examples in a worst-case fashion. In some essential way, the generation problem is therefore different from these other learning tasks in crucial ways that more detailed formalisms may potentially obscure.

Despite the generality of the model, producing the generation algorithm that proves our main theorem makes use of subtle structural properties of the given list of candidate languages. Again, we defer any detailed description to the subsequent sections, but the idea at a high level is to maintain a sequence of "provisional languages" among the candidate languages that are consistent with the finite sample $S$ from $K$ seen so far, and to continually refine this sequence of provisional languages as the adversary adds more strings to the sample $S$. Since the Gold-Angluin result says that the algorithm can never be sure which is the true language $K$, there is a sense in which this refinement process essentially needs to continue indefinitely, and in general it leads the algorithm to generate from provisional languages that may be increasingly "thin" subsets of $K$. This does not cause trouble for the specification of language generation, since it is acceptable to produce any unseen string from $K$, but it does mean that while the algorithm is able to eventually produce an infinite sequence of unseen strings from $K$, in general it might do so from a narrow part of $K$.

This property of the solution in the presence of an adversary suggests interesting connections to the problem of generation in practice as well. In particular, any method for generation has to deal with an underlying *validity problem* — producing valid outputs — and an underlying *breadth problem* — producing outputs that represent the full range of valid outputs in some reasonable way. The breadth problem is notoriously difficult, and it manifests itself in numerous ways in the methodology of machine learning and generative models. The approach that proves our main result helps illustrate the tension between validity and breadth even in settings with worst-case assumptions rather than probabilistic ones, and this tension shows up in both the early phases of our algorithm's execution and the later phases. In the early phases, before the algorithm has refined its provisional language sufficiently, it is generating too broadly and producing strings that are not part of the target language $K$ — an analogue at a high level of a kind of hallucination in which the generated strings belong to some consistent candidate language, but not to the actual target language [7, 8, 11]. In the later phases, on the other hand, the algorithm continuously shrinks its range of possible outputs so as to ensure that they will be contained within $K$ — sacrificing validity for breadth in a manner analogous to the issues that arise in the problem of mode collapse for generative models [3, 4]. Our model therefore suggests interesting questions about the fundamental trade-offs that may exist between validity and breadth even in settings without an underlying probabilistic model.

## 2   Formal Model and Results

We now provide a formal description of the model and the statement of our results. To begin with, we have a countable list of candidate languages $\mathcal{C} = \{L_1, L_2, L_3, \ldots\}$, where each $L_i$ is a subset of some countable set $U$. All we assume about the list of languages is that it is specified through a black box that can answer questions of the form "Is $w \in L_i$?" for any string $w \in U$ and language $L_i \in \mathcal{C}$. (If the reader finds it helpful for concreteness, they can consider the results that follow in the context of a specific list of languages $\mathcal{C}$, such as the set of all context-free languages or the set of all regular languages; but everything we say applies to general collections of languages.) We will allow the collection $\mathcal{C}$ to contain repetitions, in that we may have $L_i = L_j$ for different indices $i$ and $j$. We will assume that all the languages $L_i$ are infinite; while the original Gold-Angluin framework did not require this, it becomes important in specifying the generation problem: if we require an algorithm to output unseen strings forever, then this is not possible from a finite language, where the algorithm would eventually run out of new strings to generate.

An adversary and an algorithm now play the following game. The adversary chooses a language $K$ from $\mathcal{C}$ without revealing it to the algorithm, and it begins enumerating the strings of $K$ one by one over a sequence of steps $t = 1, 2, 3, \ldots$. The adversary can repeat strings in its enumeration, but crucially, for every string $w \in K$, there must be at least one time step $t$ in which $w$ appears. Let $S_t$ be the set of strings that the adversary has enumerated in steps 1 through $t$.

**Identification and Generation.**   In this framework, we can now specify both the Gold-Angluin problem of identification and the contrasting problem of generation that we study in this paper.

- *Identification (from [6, 2]:* In each step, the algorithm observes $S_t$ and must output an index $i$ (its guess for the true language $K$). The algorithm *identifies $K$ in the limit* if there is some $t^*$ such that for all steps $t \geq t^*$, the algorithm's guess in step $t$ is an index $i$ for which $L_i = K$.

- *Generation (from the present paper):* In each step, the algorithm observes $S_t$ and must output a string $a_t$ (its guess for an unseen string in $K$). The algorithm *generates from $K$ in the limit* if there is some $t^*$ such that for all steps $t \geq t^*$, the algorithm's guess $a_t$ belongs to $K - S_t$.

A key point in both problem formulations is that the algorithm is not told if its guesses are correct.

We know from the Gold-Angluin results that there is no algorithm that can achieve identification in the limit for an arbitrary countable collection $\mathcal{C}$ of languages (or even for specific countable collections, like the set of all regular languages or the set of all context-free languages). Our main result is a dramatically different answer for language generation; it is possible for every countable collection:

**(2.1)** *There is an algorithm with the property that for any countable collection of languages $\mathcal{C} = \{L_1, L_2, L_3, \ldots\}$, and any enumeration of one of these languages $K$, the algorithm generates from $K$ in the limit.*

**A Result for Finite Collections.** We prove a second result as well, focusing on the variant of the problem in which the collection of languages $\mathcal{C}$ is finite. In this case, it follows from Angluin's characterization that every finite collection $\mathcal{C}$ allows for identification in the limit. Given this, what more might we ask for? A natural question is whether there is a *uniform* bound on the number of samples needed to ensure that the algorithm can correctly identify the true language $K$: for any finite collection $\mathcal{C}$, is there a bound $t(\mathcal{C})$ and an algorithm with the property that after seeing any $t(\mathcal{C})$ distinct strings from $K$, the algorithm is guaranteed to report $K$ as its guess for the true language?

It is easy to see that for the Gold-Angluin model of learning, this is not possible. For example, suppose that $\mathcal{C}$ is the collection consisting of two languages $L_1$ and $L_2$: $L_1$ consists of all possible strings, and $L_2$ consists of all strings of even length. Suppose there were a bound $t(\mathcal{C})$ and an algorithm that was guaranteed to guess correctly after seeing $t(\mathcal{C})$ distinct samples. Then an adversary could present $t(\mathcal{C})$ distinct strings of even length, and then ask the algorithm to guess whether the true language is $L_1$ or $L_2$: if the algorithm guesses $L_2$ at this point, then the adversary could announce that the answer is $L_1$, and conversely if the algorithm guesses $L_1$. This does not prevent the algorithm from learning the true language in the limit (since the adversary must eventually output an odd-length string if the true answer is $L_1$); but there is no fixed bound $t(\mathcal{C})$ by which this can be guaranteed.

However, for the problem of generation with a finite collection of candidate languages, we can provide this much stronger type of uniform bound, via an algorithm that generates correctly after seeing a finite sample whose size $t(\mathcal{C})$ is specified at the outset. In fact, we can achieve more: after seeing this finite sample, the algorithm can generate an infinite sequence of unseen elements from the true language.

**(2.2)** *There is an algorithm with the property that for any finite collection of languages $\mathcal{C}$, there is a number $t(\mathcal{C})$, such that for any language $K$ in $\mathcal{C}$, and any sequence $S$ of at least $t(\mathcal{C})$ distinct elements from $K$, the algorithm can produce an infinite sequence of distinct strings from $K - S$.*

In subsequent work building on the present paper, Raman and Tewari [10] have recently proved further results about the type of *uniform generation* that we consider in (2.2), when it is possible to put an a priori bound $t(\mathcal{C})$ on the number of distinct strings an algorithm needs to see from $K$ before it can be guaranteed to begin generating unseen strings from $K$. They also consider variants of this definition, situating their analysis in the framework of classical learning theory.

**Extensions and Generalizations.** Following these two main results in our basic model of generation, we provide (in Section 7) the following generalization of the model. Specifically, a familiar issue from language generation applications is the role of the *prompt*: a user provides an input string $p$, and a generation algorithm is then supposed to produce a "continuation" string $c$ to come after the prompt, so that the concatenation of $p$ and $c$ is a valid utterance. We offer an extension that incorporates the notion of prompting, while maintaining the basic structure of the model, and we show how to formulate and prove a generalization of our first main result (2.1) in a setting where at each time step the adversary is allowed to specify a prompt that must be completed.

# 3 An Approach to Generation that Doesn't Work

The following section is important, because it describes an approach that is arguably the most natural non-trivial attempt at achieving language generation in the limit. It seems at first seems to solve the problem directly, but in fact it fails for a deep reason that is important to understand, since it motivates the more involved solution that follows.

The strategy is to move through the list of languages $\mathcal{C} = \{L_1, L_2, L_3, \ldots\}$ in order, treating each language $L_i$ as a hypothesis for $K$ until the sample $S_t$ proves otherwise. That is, we start with $L_1$, and we generate strings from $L_1 - S_t$ until we encounter (if ever) a step $t$ in which $S_t \nsubseteq L_1$. At this point we know that $L_1$ cannot be the true language $K$, and so we continue the process from $L_2$. The nice idea that underpins this strategy is that the true language $K$ is equal to $L_z$ for some index $z$. (Since $\mathcal{C}$ can contain repetitions, $K$ might appear several times, but we can take $L_z$ as the first appearance.) So if our process were to reach $L_z$ at some step $t^*$, it would never move on from $L_z$, and so we would be generating from $K - S_t$ for all $t \geq t^*$.

Unfortunately, there is a deep problem with this approach: there may be a language $L_i \in \mathcal{C}$ with the property that $L_i$ comes before $L_z$ and $L_i$ properly contains $L_z$ (that is, $i < z$, and $L_z \subsetneq L_i$). In this case, our procedure would stop at the first such $L_i$ forever, since it would never encounter a string in $S_t$ that didn't belong to $L_i$. And when it generated from $L_i - S_t$, there is no guarantee that it would choose strings from $L_z$.

This problem is not easily avoided, since if this approach worked as written, it would also solve identification in the limit, which we know is impossible. So we need to extract some of the useful ideas from this failed approach — in particular, the trick that $K$ appears at some finite point in the list $\mathcal{C}$, as the language $L_z$ — but add important further ideas as well. Specifically, if the algorithm is maintaining hypotheses for the true language $K$ over time, it can provably never know whether its current hypothesis is correct; instead, it must be always moving further down the collection of languages, potentially considering languages that are not $K$, but in such a way that it is eventually always generating from $K - S_t$. This is what our proof beginning in the next section will have to accomplish.

# 4 Generation in the Limit via a Function

We prove our main result in two parts. We first give a method for language generation in the limit that is not concerned with the computational power required by the agent performing the generation. Thus, rather than an algorithm to generate the string, we ask whether we can construct a function $f_{\mathcal{C}}$ based on the given language collection that maps a finite set of strings to a new string; this function takes the strings $S_t$ seen so far and outputs a string $f_{\mathcal{C}}(S_t)$ intended to be in $K - S_t$. We will prove the following:

**(4.1)** *For every countable collection of languages $\mathcal{C}$, there is a function $f_{\mathcal{C}}$ from finite subsets of $U$ to elements of $U$, such that for every enumeration of a language $K \in \mathcal{C}$, there is a $t^*$ such that for all $t \geq t^*$, we have $f_{\mathcal{C}}(S_t) \in K - S_t$.*

Note that while this positive result is not concerned with the computational power required to evaluate $f_{\mathcal{C}}$, it already contains the core contrast with language identification, which remains impossible even if we simply ask for a function $f_{\mathcal{C}}$. In the next section, we will then go on to prove (2.1) by using an algorithm that only performs standard computational steps and membership queries of the form "$w \in L_i$?"

**Minimal and critical languages.** As before, we will suppose $z$ is an index such that $L_z = K$. We say that a language $L_i$ is *consistent* with the sample at step $t$ if $S_t \subseteq L_i$. An important idea, which is implicit in our discussion of the failed approach at the end of Section 2, is that if $L_i \subseteq L_j$ are both consistent with $S_t$, then it is safer for an algorithm to generate from $L_i$ than from $L_j$: if $w \in L_j - S_t$ then we must also have $w \in L_i - S_t$. This suggests that it would be useful to find consistent languages that are *minimal* with respect to inclusion: we would say that $L$ is *minimal* if $L \in \mathcal{C}$ is consistent with $S_t$, and also $L \subseteq L'$ for every $L' \in \mathcal{C}$ that is consistent with $S_t$. Unfortunately, this is too much to ask for, since there exist instances of the problem for which there might not be any languages that are minimal with respect to inclusion. (In a finite collection of language there would need to be a minimal language, but it is easy to construct infinite collections without one.)

Therefore, we define a related concept that only involves examining the inclusion of a given language with respect to a finite set of other languages. Specifically, we look for languages $L_n$ that are consistent with $S_t$ in a given step $t$, such that $L_n$ is a subset of every consistent language that *precedes* it in the indexing of $\mathcal{C}$. We will say that such a language is *critical* at step $t$. To define this formally, we first let $\mathcal{C}_n$ denote the finite collection of languages $\{L_1, L_2, \ldots, L_n\}$. We now have the following definition.

**(4.2)**   *A language $L_n$ is* critical *at step $t$ if $L_n$ is consistent with $S_t$, and for every language $L_i \in \mathcal{C}_n$ that is consistent with $S_t$, we have $L_n \subseteq L_i$.*

**Properties of critical languages.**   At any given step $t$, there is at least one language consistent with $S_t$, since the language $L_z = K$ is always consistent with $S_t$. It follows that there is also at least one critical language at any step $t$: for any $t$, the consistent language $L_i$ with the lowest index $i$ must be critical at step $t$, as it is the only consistent language in $\mathcal{C}_i$.

Note that there can be choices of $t$ for which the language $L_z = K$ is not critical at step $t$. But a crucial fact is that $L_z$ will eventually become critical at some step $t$ and remain critical forever after that. For reasons of space, we provide complete proofs for this and all subsequent results in the Appendix, in this case in Section A.2.

**(4.3)**   *There exists a time step $t^+$ such that for all $t \geq t^+$, the language $L_z$ is critical at step $t$.*

There can be multiple critical languages at a given step $t$; for example, if on the step $t^+$ in (4.3) the first consistent language $L_i$ is not equal to $L_z$, then both $L_i$ and $L_z$ will be critical at step $t^+$. Despite the potential multiplicity of critical languages, the collection of all critical languages at step $t$ has a useful nested structure: $L_i$ and $L_j$ are both critical at step $t$, with $i < j$, then since $L_j$ is contained in all consistent languages that precede it, in particular it is contained in $L_i$. We therefore have:

**(4.4)**   *Let $i < j$, and suppose that $L_i$ and $L_j$ are both critical at step $t$. Then $L_j \subseteq L_i$.*

**A function for generation in the limit.**   At a given step $t$, suppose that the critical languages are $L_{n_1}, L_{n_2}, L_{n_3}, \ldots$ where $n_1 < n_2 < n_3 < \cdots$. (This list of critical languages might be finite or infinite.) Then (4.4) tells us that this sequence is nested by inclusion: $L_{n_1} \supseteq L_{n_2} \supseteq L_{n_3} \supseteq \cdots$.

By (4.3) we know that the language $L_z$ will eventually appear on this nested list from some step $t^+$ onward, but even then we do not know which index $n_i$ it corresponds to at any given step $t \geq t^+$. Indeed, to recall a point from earlier, the Gold-Angluin results for learning in the limit tell us that we can never know for sure which index corresponds to $L_z$. But we now arrive at the crucial point, which is that beyond some finite index, all the critical languages are subsets of $L_z$, so it is safe to generate from any of them.

Given this, we are prepared to construct our function $f_{\mathcal{C}}$.

**(4.5)**   $f_{\mathcal{C}}(S_t)$ *is defined as follows. We first identify all languages in $\mathcal{C}_t$ that are critical at step $t$. (If no such languages exist — which can only happen if none of them are consistent with $S_t$ — we define $f_{\mathcal{C}}(S_t)$ arbitrarily.) Among these critical languages, let $L_{n_t}$ be the one with the largest index $n_t \leq t$. We define $f_{\mathcal{C}}(S_t)$ to be the lowest-indexed element of $L_{n_t} - S_t$.*

Finally, to establish our initial claim (4.1), it is sufficient to prove the following (in Section A.2):

**(4.6)**   *For any language $L_z \in \mathcal{C}$ and any enumeration of $L_z$, there is a $t^*$ such that for all $t \geq t^*$, we have $f_{\mathcal{C}}(S_t) \in L_z - S_t$.*

As a final note, we observe that the current formulation of $f_{\mathcal{C}}$ allows it to generate the same string more than once, provided that this string is in $K - S_t$. However, it is not hard to modify $f_{\mathcal{C}}$ so that it generates a different string each time, essentially just by defining it so that it generates the lowest-indexed element that it hasn't already generated.

**The computational power required to produce $f_{\mathcal{C}}$.**   Our plan was to construct $f_{\mathcal{C}}$ without worrying about the computational power required to do so (and recalling that for comparison, in the corresponding problem of identification in the limit, no function $f_{\mathcal{C}}$ achieving identification could exist regardless of the computational power required to produce it). Now that we've constructed an appropriate $f_{\mathcal{C}}$, we can ask what was in fact required computationally.

In addition to standard computational steps and membership queries of the form "$w \in L_i$?", the definition of $f_{\mathcal{C}}(S_t)$ requires that we identify the critical languages in $\mathcal{C}_t$. From the definition, we can do this provided we can answer a finite number of *subset queries* of the form "$L_i \subseteq L_j$?". So an algorithm augmented with the power to perform such subset queries can perform generation in the limit.

In the next section, we will show how to remove the necessity for subset queries, so that generation in the limit can be performed by an algorithm using only standard computational steps and membership queries.

# 5 Generation in the Limit via an Algorithm

We now prove (2.1) by giving an algorithm that generates in the limit for any countable collection of languages $\mathcal{C}$, using only standard computational steps and membership queries of the form "$w \in L_i$?"

The set of possible strings $U$ can be written as $U = \{u_1, u_2, u_3, \ldots\}$, and for simplicity we will also use the language of the positive integers to describe $U$, treating $u_i$ as the number $i$. In an enumeration of the true language $L_z = K$, let the sequence of strings that are enumerated step by step be denoted $w_1, w_2, w_3, \ldots$.

**Extending definitions to finite subsets of languages** $L_i$. One important idea in designing the algorithm is to work with finite subsets of the languages in $\mathcal{C}$, gradually expanding the size of these subsets. Thus, for a language $L_i \in \mathcal{C}$ and a number $m$, we will use $L_i[m]$ to denote the finite set $L_i \cap \{1, 2, 3, \ldots, m\}$. We extend definition (4.2) of *critical languages* from the previous section to handle finite subsets.

**(5.1)** *Let $t$ and $m$ be positive integers. A language $L_n$ is $(t, m)$-critical if $L_n$ is consistent with $S_t$, and for every language $L_i \in \mathcal{C}_n$ such that $L_i$ is consistent with $S_t$, we have $L_n[m] \subseteq L_i[m]$.*

Since $L_n \subseteq L_i$ implies that $L_n[m] \subseteq L_i[m]$ for any $m \geq 1$, we have the following analogue of (4.3).

**(5.2)** *There exists a time step $t^+$ such that for all $t \geq t^+$ and all $m \geq 1$, the language $L_z$ is $(t, m)$-critical.*

The analogue of (4.4) also still holds with this definition, using the same proof.

**(5.3)** *Let $i < j$ and suppose that $L_i$ and $L_j$ are both $(t, m)$-critical. Then $L_j[m] \subseteq L_i[m]$.*

Finally, there is a basic monotonicity property of $(t, m)$-criticality that is useful to write down explicitly; we give the proof in Appendix A.3.

**(5.4)** *Suppose that $L_n$ is $(t, m)$-critical, and $m' < m$. Then $L_n$ is $(t, m')$-critical.*

## 5.1 An algorithm for generation in the limit

We now describe an algorithm for generation in the limit. As before, $S_t = \{w_1, w_2, \ldots, w_t\}$ is the subset of $K$ enumerated through step $t$, treating the $w_i$ as integers. We will consider the languages in $\mathcal{C}_t$ in step $t$, and maintain an auxiliary variable $m_t$, roughly corresponding to how large a prefix $L_i[m]$ we consider from each language $L_i \in \mathcal{C}_t$.

At the start of step $t$, we set $m_t = \max(m_{t-1}, w_t)$; note that by induction this implies $m_t \geq \max_{t' < t} w_{t'}$. (Later we will increment $m_t$, so we should think of it as a variable in the algorithm whose value can be modified.) We then determine which $L_i \in \mathcal{C}_t$ are consistent with $S_t$; note that by the definition of $m_t$, it is sufficient to perform membership queries for only the finite set of elements in $L_i[m_t]$ in order to do this. If there are no consistent languages in $\mathcal{C}_t$, then we output a string arbitrarily.

Otherwise, there is at least one language consistent with $S_t$, and so there is at least one $(t, m)$-critical language for any choice of $m$, since the first consistent language is $(t, m)$-critical for all $m$. Our goal is to imitate the plan from (4.5) and generate a new string from the highest-indexed critical language. But to do this, we have to find a new string, and this will in general require performing additional membership queries.

**Generating a string.** For any choice of $m$, let $n_t(m)$ be the maximum index of a $(t, m)$-critical language from $\mathcal{C}_t$; as noted above, $n_t(m)$ is well-defined since we are in the case where at least one language in $\mathcal{C}_t$ is consistent with $S_t$, and so the first consistent language is $(t, m)$-critical for all $m$. We now search for a string to generate as follows.

We maintain a counter $m$ that begins at $m_t$ and gets incremented iteratively, with each iteration doing the following:

- Increment $m$ by 1.

- Perform membership queries to determine $L_i[m]$ for each $L_i \in \mathcal{C}_t$. Note that since $m \geq m_t \geq \max_{t' < t} w_{t'}$, the determination of which languages in $\mathcal{C}_t$ are consistent with $S_t$ does not change when we do this.

- Determine which languages are $(t, m)$-critical, and from this determine $n_t(m)$. Note that this only requires consulting the results of membership queries already performed.

- If $u_m \in L_{n_t(m)}$, then output $u_m$ and define $m_t = m$. Otherwise, continue to the next iteration.

**Analyzing the algorithm.** As written, it is not immediately clear that the algorithm's iterations in step $t$ will come to an end with an output string $u_m$, rather than running forever. But we can show the algorithm does terminate, via the following claim proved in Section A.3.

**(5.5)** *In step $t$, the algorithm outputs a string after a finite sequence of operations.*

Given that (5.5) establishes that the algorithm outputs a string in step $t$, it is useful to record an additional property of the algorithm that follows directly from its construction.

**(5.6)** *In step $t$, there is an $m_t$ and an $n_t$ such that the algorithm outputs a string from $L_{n_t}[m_t]$, where $L_{n_t}$ is the $(t, m_t)$-critical language with maximum index in $\mathcal{C}_t$.*

Finally, we have the natural analogue of (4.6), proved in Section A.3, from which our main result (2.1) directly follows.

**(5.7)** *For any language $L_z \in \mathcal{C}$ and any enumeration of $L_z$, there is a $t^*$ such that for all $t \geq t^*$, the algorithm generates a string in $L_z - S_t$.*

Similar to the end of Section 4, it is straightforward to modify the algorithm so it generates strings without repetition.

## 6 Generation for Finite Collections of Languages

We now turn to our second main result, (2.2), which derives a stronger conclusion for finite collections of languages.

The proof of this result takes a different approach from what we used in the previous two sections; for the problem in this section, at a given step $t$, the approach we follow is to take the finite sample $S_t$ seen so far and ask whether there are any additional strings $w \notin S_t$ that are common to all the languages in $\mathcal{C}$ that are consistent with $S_t$. If so, then we can output such a string and be sure of satisfying the specification for generation.

**The closure operation.** We formalize this plan using the notion of a *closure* operation, as follows. For a sequence of strings $S_t$ from a language in $\mathcal{C}$, we define the *closure* of $S_t$ in $\mathcal{C}$, denoted $\langle S_t \rangle$, to be the intersection of all languages in $\mathcal{C}$ that are consistent with $S_t$. If there is a string in $\langle S_t \rangle - S_t$, then it is always safe for the algorithm to generate such a string; by definition, it must be an unseen string from the true language $L_z$. The closure operation would not have been useful for us in the case of general infinite collections of languages $\mathcal{C}$, since there are instances of the general problem where $\langle S_t \rangle = S_t$ and so the closure does not provide us with any proposals for new strings to generate. We give an example of such an instance in Appendix A.6.

However, in the case of finite collections $\mathcal{C}$ such as we are considering in this section, the closure operation becomes a powerful tool: the crux of proving (2.2) is to show that once $S_t$ is large enough, $\langle S_t \rangle$ must become infinite, so we will have an infinite set of options in $\langle S_t \rangle - S_t$ to use.

**Proving the result for finite collections.** We start by giving the basic idea behind the proof, and then the proof itself. Let us write the finite collection of candidate languages as $\mathcal{C} = \{L_1, L_2, \ldots, L_n\}$, and suppose that after the adversary has enumerated a set $S$ of strings, the languages consistent with $S$ are $L_{i_1}, L_{i_2}, \ldots, L_{i_k}$. Note that the true language $K$ must be one of these $k$ languages. Now, the closure $\langle S \rangle$ is equal to the mutual intersection $L_{i_1} \cap L_{i_2} \cap \cdots \cap L_{i_k}$, and there are two possibilities: either $\langle S \rangle$ is infinite, or it is finite. If it is infinite, then the algorithm can safely generate all of the strings in $\langle S \rangle - S$, and thus achieve the goal specified by (2.2). On the other hand, if $\langle S \rangle$ is finite, then it has size equal to some natural number $m$; in this case, after the adversary enumerates at most $m + 1$ more distinct strings, the algorithm will learn that at least one of $L_{i_1}, L_{i_2}, \ldots, L_{i_k}$ is no longer consistent. We will then have a set of at most $k - 1$ consistent languages, and we can iterate this argument at most $k - 2$ more times until (i) there is only a single consistent language, which must be $K$, or (ii) more generally, the set of all consistent languages has a mutual intersection that is infinite, in which case the algorithm can safely generate from this infinite set.

This argument conveys the key point underlying the proof, that as the adversary enumerates strings from $K$, it cannot prevent itself from reaching a point where the set of strings it has enumerated has an infinite closure. To turn this into an argument that produces a uniform bound on how many strings are needed before the closure must become infinite, we replace the iterative argument in the previous paragraph with one that is shorter and more direct. Specifically, consider all sub-collections of languages from $\mathcal{C}$ (where we think of a sub-collection as any way of choosing some of the languages from $\mathcal{C}$ but not others). Note that since $\mathcal{C}$ is finite, there are only finitely many possible sub-collections of languages from $\mathcal{C}$. For each sub-collection $\mathcal{C}'$, the mutual intersection of the languages in $\mathcal{C}'$ is either finite or infinite. Consider the sub-collections that have a finite mutual intersection, and let $m^*$ be the maximum size of such a mutual intersection. Now, suppose the adversary produces a set $S$ of $m^* + 1$ distinct strings from $K$. If we consider the sub-collection of all languages in $\mathcal{C}$ that are consistent with $S$, its mutual intersection must contain $S$ and therefore it has cardinality greater than $m^*$. By the definition of $m^*$, this means that its cardinality must be infinite. So the closure $\langle S \rangle$ is infinite, and therefore the algorithm can safely generate all the strings in $\langle S \rangle - S$.

We give a formal version of this argument in Section A.4, providing the complete proof of (2.2).

# 7 Extension: Prompted Generation in the Limit

As discussed in Section 2, a natural generalization is to ask whether we can preserve the basic structure of the model while adding a notion of *prompting*: the algorithm is provided with a *prompt* string and it must complete it to a valid output. The overall set-up is as before: an adversary chooses a true language $K$ from $\mathcal{C}$, and it begins enumerating the strings of $K$ in a sequence of steps.

**A model of prompting.** The new feature of the problem in our generalization is that in every step $t$, the adversary provides the algorithm with two things: a string $w_t$ from the true language $K$, and a string $p_t$ that serves as a *prompt*. (The adversary is allowed use the same prompt in multiple steps.) The algorithm in step $t$ must then produce a string $c_t$ with the goal that the concatenation of $p_t$ and $c_t$ is a string belonging to $K - S_t$; that is, it should be an unseen string from $K$. In what follows, we will use $p_t \cdot c_t$ to denote the concatenation of $p_t$ and $c_t$.

We observe that it leads to an equivalent problem whether we ask the algorithm to output $c_t$ so that $p_t \cdot c_t \in K - S_t$, or whether we ask the algorithm to output the full contatenated string $a_t = p_t \cdot c_t$. In this latter formulation, we can phrase the algorithm's task as follows: given a prompt $p_t$, output a string $a_t$ with the properties that $p_t$ is a prefix of $a_t$, and $a_t \in K - S_t$. Because it makes the exposition slightly simpler, we will take this latter formulation as our default way of describing the problem, but the two formulations are equivalent.

To establish a positive result for prompted generation in the limit, we need to impose some type of restriction on the prompts the adversary can provide. For example, if the adversary were simply to provide an arbitrary string $p_t$ and ask the algorithm if there exists a string $c_t$ and a language $L_i \in \mathcal{C}$ for which $p_t \cdot c_t \in L_i$, this is not a problem that could be solved by an algorithm that must terminate and whose only access to $\mathcal{C}$ comes in the form of membership queries of the form "Is $w \in L_i$?"

Here we describe a result based on the following restriction on the adversary's prompts: We say a prompt $p$ is *robust* if for all languages $L_i \in \mathcal{C}$, there exist arbitrarily long strings $c$ for which $p \cdot c \in L_i$. In the present discussion we will consider adversaries that only provide robust prompts.

We say that the algorithm achieves *prompted generation from $K$ in the limit* if there is some $t^*$ such that for all steps $t \geq t^*$, the algorithm's output $a_t$ has the property that $p_t$ is a prefix of $a_t$ and $a_t \in K - S_t$. We can prove the following.

**(7.1)** *There is an algorithm with the property that for any countable collection of languages $\mathcal{C} = \{L_1, L_2, L_3, \ldots\}$, and any enumeration of one of these languages $K$ accompanied by a sequence of robust prompts, the algorithm achieves prompted generation from $K$ in the limit.*

We provide a proof of this theorem in Section A.5 in the Appendix. In the full version of the paper we also explore stronger statements that are based on giving the adversary more freedom in the prompts it is allowed to provide.

We close with two initial observations about (7.1). First, (7.1) is a strict generalization of our first main result (2.1), since if the adversary always provides the empty string as its prompt $p_t$, then the problem of finding continuations $c_t$ for which $p_t \cdot c_t \in K - S_t$ is simply the problem of finding strings $c_t$ in $K - S_t$, as in the original definition of generation in the limit. Moreover, the empty string is a robust prompt, since each of the languages $L_i \in \mathcal{C}$ is infinite, and so there are arbitrarily long continuation strings that belong to $L_i$ when concatenated to the empty string.

Second, we observe that there is no requirement that the algorithm has ever seen a string beginning with the prefix $p_t$ among the adversary's examples $S_t \subseteq K$ before the first step $t$ in which the adversary provides $p_t$. An important point to notice about (7.1) is that the algorithm can achieve prompted generation in the limit despite this challenge.

# 8 Concluding Remarks

Generating from a language based on observed samples is thus a fundamentally different, more tractable problem than identifying the language from observed samples. It is so tractable, in fact, that it can be accomplished provided only that we know the samples come from a language in a known countable collection of languages.

It is therefore interesting to ask what stylized conclusions we might draw from these general results about generation as a task, and its relation to other learning processes. In the case of finite collections of languages, the basic idea underlying the proof is that a large "core" to the language (the closure of the sample, in our terminology) emerges at a known time after a finite set of observations, and it is then enough to generate from this core even though there might always remain peripheral parts of the language — disjoint from this core — that we can never be sure about. In the case of infinite collections of languages, the task is even more complex, because there is never a known time at which a core to the language emerges. Instead, the algorithm may need to continually shrink the set it is using for generation; through this infinite process of shrinkage, the algorithm can be sure that beyond a certain point, it is always generating from the true language $K$, even if it can not be sure when it has reached this point or what the true language is.

Thus, as noted earlier, the solutions we develop highlight interesting tensions between the problem of producing *valid* strings that belong to the target language, and the problem of maintaining *breadth* by not restricting to only a small subset of the target language. Our approaches achieve validity through a strategy that implicitly gives up on breadth, and it is interesting to ask if this is essentially necessary for any method that achieves language generation in the limit. In this way, our formalism may also help shed light on a particular broader impact of large language models, which is their implications for the homogenization of text style, in which writing becomes less varied; our results provide further indications that decreased breadth in text output may be a crucial feature of successful solutions to text generation.

This tension between validity and breadth, as it arises in our solution, also creates an interesting echo of the human process by which people acquire the vernacular within a new community [5]: as with our solution in this abstract model, people encountering the dialect in a new community similarly pass through a sequence of conceptually distinct phases: an initial phase in which they are generating too adventurously and producing invalid utterances; then a phase where the utterances are approximately aligned with the scope of the language; and finally a phase in which the range of utterances they are willing to generate shrinks further over their lifetime, as they become increasingly conservative in what they consider valid. Again, it is interesting to consider whether this type of structure is inherent in any solution to the task of generation in the limit.

**Acknowledgements.** We thank Bobby Kleinberg, Lillian Lee, Marios Papachristou, and Kenny Peng for helpful discussions on these questions and on early drafts of this paper. The work has been supported in part by a Vannevar Bush Faculty Fellowship, a Simons Collaboration grant, a grant from the MacArthur Foundation, and the Center for Applied AI at the University of Chicago Booth School of Business.

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

# A  Appendix

## A.1  Review of Negative Results for Identification

In the interests of completeness, we provide a self-contained proof of the Gold-Angluin result that language identification in the limit is not possible in general. adapting the exposition from [6, 9]. As noted in the main text, this proof is not necessary for the results in our paper, but it provides useful background and context for thinking about the problem.

There are many ways to show the impossibility of language generation in the limit using very simple language families, and we choose one that highlights some intuitive contrasts with generation. For the argument, let $U$ be the set of all integers, and let the collection of languages — each of which is a subset of $U$ — be the set of all infinite arithmetic progressions of integers. (The choice of which countable ground set $U$ we use is not crucial for any of these results, and examples are often easier to describe over the set of integers than over the set of finite strings.) Formally, for an arbitrary integer $a$ and a positive integer $b$, let $P_{a,b}$ be the arithmetic progression consisting of all integers of the form $\{a + bi : i = 0, 1, 2, \ldots\}$; and let $Q_{a,b}$ be the "bidirectional" arithmetic progression consisting of all integers of the form $\{a + bi : i = \ldots, -2, -1, 0, 1, 2, \ldots\}$. Let the collection $\mathcal{C}$ consist of all arithmetic progressions $P_{a,b}$ and $Q_{a,b}$.

Now, suppose by way of contradiction that there is an algorithm that can identify in the limit an adversarially chosen arithmetic progression $K \in \mathcal{C}$. We construct an enumeration of $K$ that causes the algorithm to fail, as follows. First, for integers $a \leq b$, let $I[a, b]$ be the interval of all integers $c$ for which $a \leq c \leq b$. We enumerate elements of $U$ in stages, where each stage $s \geq 0$ consists of a set of consecutive steps. If by induction stage $s$ has enumerated the elements of the interval $I[-s, j(s)]$ for some $j(s) \geq 0$, then stage $s + 1$ will enumerate additional elements so that by the end of the stage we have enumerated exactly $I[-(s + 1), j(s + 1)]$ for some $j(s + 1) > j(s)$. In particular, stage $s + 1$ first enumerates $-(s + 1)$ and $j(s) + 1$, and then it begins enumerating $j(s) + 2, j(s) + 3, \ldots$ in increasing order. At some point during this process, the algorithm must output $P_{-(s+1),1}$ as its guess for $K$, since we can continue in this way to produce a full enumeration of $P_{-(s+1),1}$, at which point the true language is $K = P_{-(s+1),1}$. Once the algorithm outputs $P_{-(s+1),1}$ as its guess during stage $s + 1$, we end stage $s + 1$, defining $j(s + 1)$ to be largest integer we've enumerated up to that point, and we begin stage $s + 2$.

In this way, the successive stages extend the interval $I[-s, j(s)]$ unboundedly in both directions. We are therefore enumerating $Q_{0,1} = U$, and so in fact $K = Q_{0,1}$. But we have also produced an unbounded sequence of steps $t_0 < t_1 < t_2 < \cdots$ such that the algorithm guesses $P_{-j,1}$ at step $t_j$. Thus, there is no time step $t^*$ for which the algorithm outputs the (correct) guess $Q_{0,1}$ at every $t \geq t^*$.

As a final note, we observe that for this simple collection of languages, the problem of generation in the limit is straightforward: once the algorithm has seen two elements $i$ and $j$ from $K$, with $i < j$, it knows from the arithmetic progression structure that by setting $d = j - i$, it can safely output $j + d, j + 2d, j + 3d, \ldots$, and all of these will belong to $K$. The generation task for this language family is thus particularly simple (for others it is much more subtle, as in the example from Appendix A.6), but this contrast with the negative result for identification provides some initial intuition about the differences between the two problems that we will need to exploit in a generation algorithm.

## A.2  Proofs from Section 4

In the next three subsections of the Appendix, we provide proofs of the results stated in the main text.

*Proof of (4.3).*  Let $K$ be the indices $i < z$ for which $L_z \nsubseteq L_i$. For each $i \in K$, let $v_i$ be an element of $L_z - L_i$. Let $t_i$ be the step in which $v_i$ first appears in the enumeration of $L_z$, and let $t^+ = \max_{i \in K} t_i$.

Now, suppose by way of contradiction that for some $t \geq t^+$, the language $L_z$ is not critical at step $t$. In this case, there must be some $L_i \in \mathcal{C}_z$ such that $L_i$ is consistent with $S_t$ and $L_z \nsubseteq L_i$. But we know that $v_i \in S_t$ and $v_i \notin L_i$, contradicting the consistency of $L_i$ with $S_t$.  □

*Proof of (4.6).*  In the given enumeration of $L_z$, (4.3) tells us that there will come a step $t^+$ such that for all $t \geq t^+$, the language $L_z$ is critical at step $t$. Let $t^* = \max(z, t^+)$. In every step $t \geq t^*$,

our construction of $f_{\mathcal{C}}$ will include $L_z$ among its critical languages in $\mathcal{C}_t$. Therefore, the highest-indexed critical language $L_{n_t} \in \mathcal{C}_t$ satisfies $n_t \geq z$, and so by (4.4) we have $L_{n_t} \subseteq L_z$. Since $f_{\mathcal{C}}(S_t) \in L_{n_t} - S_t$, we have $f_{\mathcal{C}}(S_t) \in L_z - S_t$ as required. $\qquad\square$

## A.3  Proofs from Section 5

*Proof of (5.4).* Since $L_n$ is $(t, m)$-critical, we know it is consistent with $S_t$, and that $L_n[m] \subseteq L_i[m]$ for all languages $L_i \in \mathcal{C}_n$ such that $L_i$ is consistent with $S_t$. Now, if $L_i$ is a language in $\mathcal{C}_n$ that is consistent with $S_t$, then since $L_n[m] \subseteq L_i[m]$ and $m' < m$, we have $L_n[m'] \subseteq L_i[m']$. It follows that $L_n[m'] \subseteq L_i[m']$ for all languages $L_i \in \mathcal{C}_n$ such that $L_i$ is consistent with $S_t$, and so $L_n$ is $(t, m')$-critical. $\qquad\square$

*Proof of (5.5).* We identify each iteration with the value of $m$ after the initial increment of the iteration; so the iterations begin at $m_t + 1$ and continue upward from there. Suppose by way of contradiction that the algorithm performs an infinite sequence of iterations.

Let us call an iteration $m$ *disruptive* if $n_t(m) \neq n_t(m - 1)$. Since $n_t(m - 1)$ is the maximum index of a $(t, m - 1)$-critical language, and since our monotonicity property (5.4) implies that $L_{n_t(m)}$ is also $(t, m - 1)$-critical, it follows that $n_t(m) < n_t(m - 1)$. Since $n_t(m)$ starts at a value bounded by $t$ and decreases by at least one with every disruptive iteration, there can be at most $t - 1$ disruptive iterations.

The sequence of iterations must therefore contain a last disruptive iteration $m^*$. For all iterations $m > m^*$, the language $L_{n_t(m)}$ does not change. Since the language is infinite, we must eventually reach an iteration $m > m^*$ for which $u_m \in L_{n_t(m)}$, and the algorithm will stop and output $u_m$ at this point. $\qquad\square$

*Proof of (5.7).* In the given enumeration of $L_z$, (5.2) tells us that there is a $t^+$ such that for all $t \geq t^+$ and all $m \geq 1$, the language $L_z$ is $(t, m)$-critical. Let $t^* = \max(z, t^+)$. In every step $t \geq t^*$, by (5.6) there is an $m_t$ such that the algorithm generates a string from $L_{n_t}[m_t]$, where $L_{n_t}$ is the $(t, m_t)$-critical language with maximum index in $\mathcal{C}_t$. In each such step $t$, $L_z$ is a $(t, m_t)$-critical language in $\mathcal{C}_t$, and so $n_t \geq z$. From (5.3), it follows that $L_{n_t}[m_t] \subseteq L_z[m_t] \subseteq L_z$. Since the algorithm's output comes from $L_{n_t}[m_t] - S_t$, it follows that it comes from $L_z - S_t$ as well. $\qquad\square$

## A.4  Proofs from Section 6

*Proof of (2.2).* We begin with some additional definitions. For any subset $A$ of the indices $\{1, 2, \ldots, n\}$, let $D(A)$ be the intersection of the languages whose indices are in $A$; in other words, $D(A) = \bigcap_{i \in A} L_i$. For any sequence $S$ of strings from a language in $\mathcal{C}$, let $I(S)$ be the set of indices of the languages in $\mathcal{C}$ that contain $S$; that is, $I(S) = \{i : S \subseteq L_i\}$. We observe that the closure operator can be written in terms of this notation, in that $\langle S \rangle = D(I(S))$.

If $D = D(\{1, 2, \ldots, n\})$ is infinite, then the algorithm can generate arbitrary strings from $D$ as its output without seeing any sample of strings at all; since $D \subseteq L_i$ for every language $L_i \in \mathcal{C}$, in particular $D \subseteq L_z$ for the true language $L_z$, and this satisfies the requirements of (2.2).

For the rest of the proof, we therefore suppose $D(\{1, 2, \ldots, n\})$ is finite. Let $\mathcal{F}$ be the collection of all sets of indices $A \subseteq \{1, 2, \ldots, n\}$ with $D(A)$ finite. Finally, let $m^* = \max_{A \in \mathcal{F}} |D(A)|$; since $|\mathcal{F}| \leq 2^n$, we observe that $m^*$ is the maximum of a finite set of positive integers, and hence a positive integer. We now define $t(\mathcal{C}) = m^* + 1$ and claim that this choice of $t(\mathcal{C})$ satisfies the required guarantee of (2.2). Indeed, consider the true language $L_z \in \mathcal{C}$ and any sequence $S_t$ of $t(\mathcal{C})$ distinct elements from $L_z$. Recall that $I(S_t)$ denotes the set of indices of all languages in $\mathcal{C}$ that contain $S_t$. We have $S_t \subseteq \langle S_t \rangle = D(I(S_t))$. If $D(I(S_t))$ were finite, then by the definition of $m^*$, the cardinality of $D(I(S_t))$ would be at most $m^*$. But this would contradict the fact that $D(I(S_t))$ contains $S_t$, which has cardinality $m^* + 1$.

Therefore $\langle S_t \rangle = D(I(S_t))$ is infinite, and it is a subset of the true language $L_z$. To conclude the proof, we therefore need only show that there is an algorithm that can enumerate all of $\langle S_t \rangle - S_t$

using only membership queries. To do this, the algorithm begins by querying whether each $w_i \in S_t$ belongs to each $L_j \in \mathcal{C}$. From this, it can determine the set $I(S_t)$ of indices of languages that contain $S_t$. Now, it enumerates every string $u_i \in U$ in ascending order, skipping the elements of $S_t$. For each such string $u_i$, it queries whether $u_i \in L_j$ for each $j \in I(S_t)$, and it outputs $u_i$ if it belongs to each of these languages. In this way, the algorithm enumerates the infinite set $\langle S_t \rangle - S_t \subseteq L_z - S_t$ after seeing a sample of $t(\mathcal{C})$ strings in $L_z$. $\qquad \square$

### A.5 Proofs from Section 7

We now describe how to prove our result (7.1) on prompted generation with robust prompts. The proof is a direct adaptation of the proof of (2.1) from Section 5; as we will see, the structure of critical languages built up there is sufficiently strong that not much more is needed to handle the prompted version of the problem with robust prompts.

As in Section 5, we will work with a specific enumeration $u_1, u_2, u_3, \ldots$ of all strings in $U$, and work with finite subsets of the languages $L_i$, defined via the notation $L_i[m] = L_i \cap \{u_1, u_2, \ldots, u_m\}$. The algorithm for prompted generation will closely follow the algorithm from Section 5, in that in every step $t$, it will increment a counter $m$ and maintain knowledge of the maximum index $n_t(m)$ of a $(t, m)$-critical language from $\mathcal{C}_t$. Maintaining knowledge of $n_t(m)$ does not require knowledge of the prompts, and so this part of the algorithm is the same as before. What changes is the stopping condition for the algorithm in step $t$: rather than continue increasing $m$ until any valid output is found — that is, until $u_m \in L_{n_t(m)}$ — the algorithm must increase $m$ potentially even further, until it finds a string $u_m$ for which $p_t$ is a prefix of $u_m$, and $u_m \in L_{n_t(m)} - S_t$. However, since $p_t$ is a robust prompt, the algorithm is guaranteed to eventually find such a string, and so we can be sure that its iterations in step $t$ will terminate. If we let $m_t$ be the value of $m$ at the end of step $t$, then once $t$ is large enough, we know that $L_{n_t(m_t)}[m_t] \subseteq L_z[m_t]$, where $L_z = K$ is the true language, and so the string $u_m$ that it outputs has $p_t$ as a prefix and belongs to $K - S_t$ as required.

The discussion above provides the entire set of modifications to the algorithm; for completeness we now describe these in more detail, together with a proof of correctness.

First, the facts (5.2) through (5.4) still hold in the prompted case, since they are structural properties of the language that are not affected by the adversary's use of prompts. The algorithm for generating an output string uses an iteration in step $t$ for which parts (i), (ii), and (iii) of each iteration are the same as in Section 5. Step (iv) of each iteration is replaced by

(iv′) If there is any string $u_i$ for $i \le m$ such that $u_i$ has $p_t$ as a prefix and $u_i \in L_{n_t(m)} - S_t$, then choose the minimum $i$ with this property; output the string $u_i$ and define $m_t = m$. If there is no such $u_i$, then continue to the next iteration.

Now, the proof of termination works as before, by establishing that there are only finitely many *disruptive iterations* in which the identity of $n_t(m)$ changes; this part does not depend on the structure of prompts but only on the definition of a $(t, m)$-critical language, and so it uses (5.2) through (5.4) exactly as before. After the last disruptive iteration, either there is a string $u_i \in L_{n_t(m)} - S_t$ with $i \le m$ for which $p_t$ is a prefix, or else the algorithm will eventually reach one, since the prompt $p_t$ is robust. It declares this $u_m$ to be its output string. We therefore have

**(A.1)** *In step $t$, if at least one language in $\mathcal{C}_t$ is consistent with $S_t$, then there is an $m_t$ and an $n_t$ such that the algorithm terminates with a string $a_t$ for which $p_t$ is a prefix of $a_t$ and $a_t \in L_{n_t(m_t)}[m_t] - S_t$, where $L_{n_t(m_t)}$ is the $(t, m_t)$-critical language with maximum index in $\mathcal{C}_t$.*

Finally, we establish the basic correctness property of the algorithm, from which (7.1) follows directly.

**(A.2)** *For any language $L_z \in \mathcal{C}$ and any enumeration of $L_z$ with robust prompts $p_1, p_2, p_3, \ldots$, there is a $t^*$ such that for all $t \ge t^*$, the algorithm generates a string $a_t$ for which $p_t$ is a prefix of $a_t$ and $a_t \in L_z - S_t$.*

*Proof.* In the given enumeration of $L_z$, (5.2) tells us that there is a $t^+$ such that for all $t \ge t^+$ and all $m \ge 1$, the language $L_z$ is $(t, m)$-critical. Let $t^* = \max(z, t^+)$. In every step $t \ge t^*$, by (A.1) there is an $m_t$ such that the algorithm generates a string $a_t$ such that $p_t$ is a prefix of $a_t$, and $a_t \in L_{n_t(m_t)}[m_t] - S_t$, where $L_{n_t(m_t)}$ is the $(t, m_t)$-critical language with maximum index in $\mathcal{C}_t$. In each such step $t$, $L_z$ is a $(t, m_t)$-critical language in $\mathcal{C}_t$, and so $n_t(m_t) \ge z$. From (5.3), it follows

that $L_{n_t(m_t)}[m_t] \subseteq L_z[m_t] \subseteq L_z$. Since $a_t \in L_{n_t(m_t)}[m_t] - S_t$, it follows that $a_t \in L_z - S_t$ as well. $\qquad\qquad\qquad\qquad\qquad\qquad\qquad\qquad\qquad\qquad\qquad\qquad\qquad\qquad\qquad\quad\;\;\Box$

## A.6  An Example Where Closure is Not Helpful

In Section 6, we introduced the notion of *closure*: For a sequence of strings $S_t$ from a language in $\mathcal{C}$, the *closure* of $S_t$ in $\mathcal{C}$, denoted $\langle S_t \rangle$, is the intersection of all languages in $\mathcal{C}$ that are consistent with $S_t$.

We observed in Section 6 that in any step $t$ where $\langle S_t \rangle$ contains an element not in $S_t$, the algorithm can always safely output an element in $\langle S_t \rangle - S_t$ and be sure that it is outputting a new element in the true language $K$ This is simply because for every consistent language $L_i \in \mathcal{C}$, we have $\langle S_t \rangle - S_t \subseteq L_i - S_t$ by the definition of the closure operation, and in particular this holds for the true language $K$.

This strategy is key to the proof in Section 6, when we were dealing with finite collections $\mathcal{C}$, and it would have been an effective strategy in the case of infinite collections $\mathcal{C}$ as well provided we could be sure that $\langle S_t \rangle - S_t$ was always non-empty. Unfortunately, there are instances of the problem with infinite collections $\mathcal{C}$ for which $\langle S_t \rangle - S_t$ is empty for arbitrarily large $t$, and therefore the closure provides us with no new strings to generate.

To see how this can happen in an example, we go back to the collection of arithmetic progresions over the integers as the ground set $U$, and we consider a more complicated collection $\mathcal{C}$ that builds on this. In particular, we let $P_{a,b}$ as before be the arithmetic progression consisting of all integers of the form $\{a + bi : i = 0, 1, 2, \ldots\}$; and now, for any finite set of integers $V$, we define the language $L(a, b, V) = P_{a,b} \cup V$. Our collection $\mathcal{C}$ consists of every $L(a, b, V)$ for an arbitrary integer $a$, an arbitrary positive integer $b$, and an arbitrary finite set of integers $V$. (Think of $L(a, b, V)$ as a copy of the arithmetic progression $P_{a,b}$ that has been obscured by an arbitrarily large finite set $V$ so that its structure is harder to discern.)

Now, suppose the adversary is enumerating a language $K = L(a, b, V) \in \mathcal{C}$, and consider the set $S_t$ of samples after $t$ steps. We claim that $\langle S_t \rangle = S_t$. Intuitively, this is because $S_t$ might have come completely from the finite set $V$ that is part of $L(a, b, V)$; more formally, there cannot be any element $j \in \langle S_t \rangle - S_t$ because $L(j + 1, 1, S_t)$ is consistent with $S_t$, and it does not contains $j$.

This example illustrates the sense in which we mean that closure by itself is insufficient to provide a strategy for generation in the limit when the collection of languages $\mathcal{C}$ is infinite: in this instance, the algorithm has to repeatedly output elements outside the closure of $S_t$ and yet somehow eventually generate from $K$. We know that the algorithm that proves our main result (2.1) is able to do this, but it is also interesting to ask if there is a simpler way to directly describe a method for generation in the limit for this particular example. In fact, there is a direct solution to this example, as follows: in any step $t$, the algorithm finds the two largest elements $i < j$ in $S_t$, and with $b = j - i$, it outputs $j + b$. The point is that if the true language $K$ is $L(a, b, V)$, there will come a time when the adversary has enumerated every element of $V$, and within a finite number of steps after this, the two largest elements of $S_t$ will have to come from $P_{a,b}$. From this step onward, the algorithm's strategy of outputting $j + b$ is guaranteed to produce an element in $K - S_t$. This particular strategy of course seems highly specific to the particular example we are discussing here, but at a very high level it does contain reflections of some of the ideas we saw in the general solution from Sections 4 and 5.

