# OpenReview forum: "Language Generation in the Limit"
_NeurIPS.cc/2024/Conference — NeurIPS 2024 spotlight_

### Official Review · Reviewer_8y2i · 2024-07-11

**Soundness:** 4
**Presentation:** 2
**Contribution:** 2
**Rating:** 5
**Confidence:** 1

**Summary:**

This paper introduces a new theoretical framework for language generation

**Strengths:**

Novel theoretical perspective

**Weaknesses:**

hard to judge

**Questions:**

hard to judge

**Limitations:**

hard to judge

---

> ### Author Rebuttal · Authors · 2024-08-06
>
> We would be happy to answer any questions you have about the submission. Given that the current review says, "hard to judge" as the full reply for the Weaknesses, Questions, and Limitations, we do not currently have enough information to proactively provide more, but we would be able to share more information if you were able to add questions to your review.

---

### Official Review · Reviewer_v4Eq · 2024-07-12

**Soundness:** 2
**Presentation:** 2
**Contribution:** 2
**Rating:** 4
**Confidence:** 4

**Summary:**

In the classical model of language learning in the limit proposed by Gold and Angluin, there is a countable sequence of candidate languages L_1,L_2, ... and a language L* that is equal to L_i for some unknown i. At each time step t, Player 1 draws a string w_t from L*, and Player 2 is required to guess a number i. The language L* is said to be learned in the limit if there is some t* such that for every t>=t*, the language L_i is equal to L*. When stated in this general form, learning in the limit is an impossible task, unless the list of languages satisfies some strong properties.

In this work, the authors propose a similar approach in the case of language generation. At each time step t, Player 1 draws a string w_t from L*, and Player 2 is required to guess a string w in L* - {w_1,...,w_t}. That is, instead of identifying the language L*, all that is required is to generate a string w that belongs to the language L* but not to the list of examples provided by Player 1. The authors claim that in this setting, generation in the limit is always possible, no matter what sequence of languages L_1,L_2,...,

**Strengths:**

The result, although being of a purely theoretical nature, may have applications to prove further results in learning theory.

**Weaknesses:**

The explanation of the key ideas of the paper seem to be much longer than necessary. It may be the case that the paper is more adequate to a conference specialized in computational learning theory, such as COLT.

I reviewed this paper before. As far as I can see, I could not find discussions in this version that address issues pointed in my previous review. So, at this point in time, I'm only able to suggest the same rating as before (weak reject). Depending on the answers from the authors I may increase my score slightly.

-------------------

 If Player 1, can ask questions of the form w \in L_i then the strategy for generating in the limit seems to be quite simple.

1) First, at Step t, Player 1 keeps track of all languages up to language L_t that are consistent with S_t. This can be done by using the membership oracle.

2) Once this has been done, Player 1 enumerates the strings in {0,1}^* \backslash S_t in order of length. For each of these strings w, Player 1 checks whether w belongs to the first consistent language in the list. If yes answer w. If no proceed to the next string in the enumeration.

This algorithm always generates some w in a finite amount of time, because the languages are assumed to be infinite. In particular, the first consistent language is infinite, and therefore, at some finite length, it will have strings of that length that does not belong to S_t. On the other hand, at some point in time (which may be unknown), the target language K will be the first language in the list. This is because of the assumption that for each w in K there is a t such that w belongs to S_t. In particular, at some finite point, none of the languages that precedes K in the list will be consistent with S_t. Additionally, by assumption K is always consistent with S_t.

Therefore, essentially, the core of the argument for proving theorem (2.1) is in the discussion of statement (4.3) in Section 4, and the membership test is used to implement this argument in practice. The argument mentioned above also gives Theorem (2.2) immediately. Just let t(C) be the number of strings of length at most r, where r is the smallest number such that each language preceding K in the list has a string u of length at most r that does not belong to K. At this point, K is guaranteed to have become the first consistent language in the list, and anything generated by the algorithm will belong to K.

I still think that the main claims of the paper are interesting from a conceptual point of view because they sound counter-intuitive at first glance. Nevertheless, at this current point in time the paper has two main drawbacks. First, it much longer than necessary when it comes to explaining the main ideas to prove 2.1 and 2.2. The argument 4.3 could be made directly after 2.1 and the discussion about membership test as synthesized above is also straightforward. The second drawback is that the paper does not provide any "application" for their result. Please note that while the bout t(C) is finite, from what I could understand, the authors provide no bound on the time necessary to generate the next string (and this may not be possible at all). Therefore, from an algorithmic point of view, the contribution of the paper is weak. The algorithm generates the next string in the list at some point in time which seems to be completely unbounded.

Please also note that the discussion in 125-129 is a bit fallacious. Of course, if the target language is finite, then the algorithm would run out of strings to generate. So it is indeed natural to assume that the target language is infinite. However, the paper is assuming that all languages in the list are infinite. This is a very strong assumption. My impression is that this assumption is only being made so that the algorithm described above (item 2) stops in finite time. If the first language in the list of consistent languages were exactly S_t, then the the enumeration process in item 2 would run forever. So, in my opinion, it is important to highlight that the assumption that all languages are infinite is being made so that the proof goes through.

**Questions:**

Please address the comments above.

**Limitations:**

Yes

---

> ### Author Rebuttal · Authors · 2024-08-06
>
> Thanks for your review; we would very much like to discuss these points with you, because the simpler argument you propose for the main result --- generation in the limit --- is not correct. Furthermore, our submitted paper contains a description of the approach you describe and an explanation for why it does not work (in Section 3, entitled "An Approach to Generation that Doesn’t Work"). Based on the earlier review you mention, we gave that discussion a prominent place in our current submission; this is a change to the current submission in response to your earlier review.
>
> We encourage you to look at this section, but we also summarize here why your simpler argument is not correct. We agree with the portion of the argument in which you describe how to generate a new string not belonging to $S_t$ from the first language in the list of all languages that are consistent with $S_t$. The problem arises with your claim that, "at some point in time (which may be unknown), the target language K will be the first language in the list. This is because of the assumption that for each w in K there is a t such that w belongs to S_t. In particular, at some finite point, none of the languages that precedes K in the list will be consistent with S_t."
>
> To see why this argument is not correct, consider (as we discuss in Section 3) the case in which $K = L_z$ for some index $z$, and there is an earlier language $L_i$ that comes before $K$ (i.e. $i < z$) such that $L_i$ is a proper superset of $K$.  (That is, we are supposing $L_i$ contains all of $K$ as well as strings not in $K$.) In this case, for every time step $t$, every string in $S_t$ must belong to $L_i$, because every string in $S_t$ belongs to $K$ by definition, and $L_i$ is a superset of $K$. Therefore, $L_i$ will always be consistent with $S_t$, for all $t$, and $K$ will never become the first consistent language in the list (because $L_i$ precedes $K$ and is always consistent). And this means that an algorithm that simply generates a string from the first consistent language on the list has no way to avoid generating a string in $L_i - K$ infinitely often, which means that the algorithm is not successfully generating from $K$ in the limit.
>
> There is a concrete sense in which there cannot be a simple "fix" to this argument that might get it to work. The reason is that if we could be sure that after a finite time $K$ was the first consistent language on the list, then we could solve the problem of language identification in the limit, not just language generation in the limit, by simply guessing that the identity of the adversary's language is the first consistent language in the list. But this would contradict Gold's Theorem, that language identification in the limit is not possible.
>
> It is because this approach cannot work that we are forced to embark on the more involved proof that requires the definition of critical languages. And once we do that, we require the further arguments in Section 5, because determining whether a language is critical (according to the definition we introduce) is not algorithmically decidable. As a result, we need -- in Section 5 -- to work with a weakened version of criticality that is both decidable and also sufficient for generation in the limit.
>
> We very much hope you go through these arguments, because they should show you why your claim about a simpler proof is not correct, and why small variations on this attempt at a simpler proof will not work (because they would provide a solution to language identification in the limit, which is not possible by Gold's Theorem). You can also see the discussion in Section 3 of the paper, which addressed these points in the submission.
>
> On the other points in your review, we agree that the presence of arbitrarily large "gaps" in some of the languages $L_i$ --- that is, arbitrarily large intervals $[a,b]$ such that $L_i$ contains no strings of length between $a$ and $b$ --- means that we cannot put an upper bound on how long (in terms of basic computational steps) the algorithm needs to generates a next string. As noted in the paper, the key issue of interest to us in this paper is to show that certain tasks are possible at all. However, we do note (as we also mention in replying to Reviewer 4Kah) that many of the cases of interest, dating back to Gold and Angluin's work, are settings in which the languages $L_i$ are regular or context-free. In these cases, the relevant pumping lemmas for these language families constrain the kinds of intervals of lengths $[a,b]$ for which a language can contain no strings of that length. This suggests the potential to extend our work to address interesting further questions of a more quantitative nature for these families of languages, and we would note these further directions in a revised version of the paper.
>
> We also agree that if some of the languages $L_i$ are finite, then an algorithm cannot know at any given time step $t$ if $L_i - S_t$ is non-empty; i.e., it cannot know if there are more strings in $L_i$ left to generate. We can note this in a revised version of the paper. We also observe, however, that the challenges in language generation tend to arise much more from uncertainty about the nature of the true language than about the risk of "running out" of strings to generate; there are for example many informal arguments suggesting why natural languages will contain arbitrarily long sentences. We believe that the assumption that the languages $L_i$ are infinite captures this point.
>
> You mentioned in your review that you were prepared to raise your score based on our response. Please let us know if we can clarify anything in the above discussion, and given our points about why simpler proofs for the main result do not appear to work, we would appreciate if you were willing to raise your score.

---

> ### Comment · Area_Chair_2j6n · 2024-08-07
>
> Dear Reviewer: please see a comment from the authors to me addressing your review:
>
> Comment:
> The instructions for the use of official comments suggest that we should write an official comment about reviews that contain inaccuracies about the submission. As such, we would like call your attention to significant technical errors in the review of our paper by Reviewer v4Eq.
>
> The bulk of Reviewer v4Eq's review is a sketch of a proof of our main result that, if correct, would be simpler than the proof in our paper. The reviewer's proof, however, is not correct, and it is based on an approach that --- in a concrete sense --- cannot work, because it would also prove a stronger result that is known to be false.
>
> Reviewer v4Eq notes that they reviewed our submission for an earlier conference. For that conference, they proposed the same simple, incorrect proof in a review comment written after the conference's rebuttal period had ended, and so we were not able to point out the error in their proposed proof. As a result, we included a section (Section 3) in the main text of our current submission that describes their proposed proof and explains why it does not work.
>
> In this way, not only is Reviewer v4Eq incorrect in their proposed proof, they are also incorrect in their point that the current submission does not address their earlier review. Rather, we highlighted the error in their proposed proof in a full section of the current submission. We did this in case we got a reviewer who made a similar error; as it turns out, we got the same reviewer, but this reviewer appears not to have seen the section that explains the error they are making.
>
> We are including a discussion of the reviewer's error in our response to them on OpenReview, but given the reviewer's persistence in making this error we wanted to include an official comment as well; we are not sure what else we can do given that we already devoted a section of the submission to alert the reviewer, unsuccessfully as it turns out, to the error they are making.

---

> > ### Comment · Area_Chair_2j6n · 2024-08-07
> >
> > Dear Authors,
> > I forwarded your comment to the reviewer.
> > Best regards,

---

> ### Comment · Reviewer_v4Eq · 2024-08-13
> **Reply Part 1**
>
> Dear Authors,
>
> thank you very much for the reply, and for adding a discussion explaining why the most straightforward approach does not work. The problem is that there can be some language with a smaller index which strictly contains the target language.
>
> Now, to avoid misunderstandings, let me state that my low score on the paper is grounded on four main reasons, which are detailed below.
>
> ------------------------------
> REASON 1) The theorem does indeed have a very short proof. Please note that while the naive approach described in my previous review does not work as pointed out by the authors, the argument has indeed a direct fix using a variant of the notion of criticality used defined by the authors. Lets call it simple-criticality.
>
> Definition (Simple Criticality): Let L_1,...,L_t be the first t languages in the list of languages, we say that a language L_j with j<=t is simply-critical if L_j is consistent with S_t and L_j\subset L_i for every other i<=t such that S_i is consistent with S_t.
>
> Generation in the Limit with Inclusion Queries and Membership Queries.
>
> 1) At each step t, Let C_t be the set of all languages up to L_t that are simply-critical for t. The construction of C_t can be realized with inclusion queries.
>
> 2) If C_t is empty, output any string. Otherwise, let L_x be the language in C_t with the smallest index. In this case, output the smallest (and lexicographically first) string w in Lx - S_t as an answer.
>
> Proof of correctness: Let L_z be the target language from which the samples S_t are observed. Since L_z is consistent with S_t for every t, we have that for any t\geq z, there is at least one simply-critical language. Now, let t\geq z. Then any language L_j that is simply-critical for t is contained in L_z. Otherwise, this would contradict minimality. Since L_j is infinite, L_j-S_t is non-empty. Therefore the string w is well defined and can be obtained with membership queries only. Since L_j \subsetq L_z we have that w\in L_z as required. QED.
>
> The inclusion queries in Item 1 can be replaced by membership tests (as done by the authors) simply by considering slices of the languages up to L_t containing only strings up to a certain length. Now the "trap" of language identification is avoided with this approach because, for each t>z, the chosen language L_x may be strictly contained in L_z. Additionally there may be an infinite chain of languages where one is a superset of the next.
>
> Please note that the own proof by the authors can be simplified significantly by using their own definition of criticality provided the statements are presented in a more natural order (as suggested in my previous review). More specifically,
>
> Statement of Theorem 2.1
> Definition of Criticality
> Proof with Inclusion tests
> Argument to replace inclusion tests by membership tests.
>
> In this sense, I do consider that the paper has a presentation problem because at its current stage it is difficult to read due to the unnecessary length of the explanations used in the proof. A simple and direct formal argumentation would be preferable to check correctness as opposed to length discussions involving intuitive, but imprecise, concepts. The space used for the proof could be used to provide additional results, such as exploring consequences of the main theorem. See Below.
>
> -----------------------------------
>
> REASON 2) The paper does not describe concrete applications of the language generation problem. This is what I mean when I write "The second drawback is that the paper does not provide any "application" for their result." There are two lines of results that I would expect to be discussed in such a paper.
>
> 2a) First, Language Identification is an important primitive in computational learning theory. There are certainly several problems of theoretical/practical interest that reduce to Language Identification. What are the analogs of these problems in the context of Language Generation? The caveat is that while reducing a problem to language identification does not help in general, the analog problems in the context of language generation would have a solution.
>
> 2b) Second, since you are selling the paper in the context of LLMs, I would expect a more specialized discussion of results specialized to the realm of LLMs. For example, what models of computation would be used to instantiate the languages L_1, L_2,... in the context of LLMs? What consequences could you derive from this specialization? Could you establish complexity bounds on the process of generating the next token (word)? Who is the generator of the set S_t? There are many more questions that are unanswered here. So In my opinion, the use of LLMs to justify the applicability of the result is very handwaived in this current version.

---

> > ### Comment · Reviewer_v4Eq · 2024-08-13
> > **Reply Part 2**
> >
> > -----------------------------------
> >
> > REASON 3) As it is currently stated, the result is a pure theoretical result in the realm of computational learning theory given that in general it is not possible to establish an upper bound on the time necessary to generate the next token. Therefore, the result is a result about "computability" not a result about "algorithmics" and much less neural processing. This is what I mean when I write: "Therefore, from an algorithmic point of view, the contribution of the paper is weak." and "It may be the case that the paper is more adequate to a conference specialized in computational learning theory, such as COLT.". In my opinion, the contributions of the paper do not have much to do with topics covered within NeurIPS.
> >
> > -----------------------------------
> >
> > REASON 4) The results of the paper only seem to hold under the assumption that all languages in the list are infinite. This seems to be an unnatural assumption. From what I understand, the proof breaks if this assumption is removed, and therefore, this leads me to conclude that the assumption is made for the sake of making the proof carry over. Please note that the authors write in line 126:
> >
> > "We  will assume that all the languages Li are infinite; while the original Gold-Angluin framework did not require this, it becomes important in specifying the generation problem: if we require an algorithm to output unseen strings forever, then this is not possible from a finite language, where the algorithm  would eventually run out of new strings to generate."
> >
> > I do not agree with this explanation. There is no apparent justification for requiring that all languages in the countable list are infinite, other than to make the proof of the main theorem work. This seems to be a very restrictive assumption, because it rules out the possibility of instantiating the result in a concrete way with any model of computation where language finitetess is undecidable. Please note that only very restricted classes of languages are known to have decidable finiteness. Going a bit beyond context-freeness already renders the finiteness test or even (emptiness) undecidable. So, this rules out the possibility of enumerating over these languages by enumerating the "machines" representing the languages.
> >
> > I believe that in your reply you agree that assuming finiteness is a drawback. Why not make this explicitly in the paper?
> >
> > -----------------------------------
> >
> > For the reasons mentioned above, I will keep my score "weak reject" mostly because I believe that presentation of the paper can be significantly improved towards clarity, and also because the results in the paper are much more in the realm of computability theory than in the realm of neural networks.

---

> > > ### Author Response · Authors · 2024-08-13
> > >
> > > Thank you for your reply, and for suggesting a fix to your earlier incorrect argument. We'd like to start by pointing out that your new proposed proof is also incorrect. The problem is in the step where you claim that "Since $L_z$ is consistent with $S_t$ for every $t$, we have that for any $t\geq z$, there is at least one simply-critical language." Here is an example that shows there might be no simply-critical language at certain steps where $t \geq z$, contradicting this claim. For the example, let the languages be subsets of the natural numbers, let $L_1$ consist of all multiples of 6, $L_2$ consist of all multiples of 10, and $L_3$ consist of all multiples of 15. (It will not be important for this example what $L_4, L_5, ...$ are.) Let $L_3$ be the true language; i.e. $z = 3$. Suppose that the adversary's first three examples are 60, 120, and 180, so at $t = 3$, the set $S_t$ is {60,120,180}.
> > >
> > > For this value $t = 3$, which satisfies $t \geq z$, each of $L_1$, $L_2$, and $L_3$ is consistent with $S_t$, but none is a subset of any of the others, so there is no simply-critical language for $t = 3$. (Note that in comparison, there is a language in this example that is critical under our definition, since as noted in the paper, the first consistent language is always critical.)
> > >
> > > Since the author response period is closing today and we may not get a chance to respond further, we would like to make a few further points based on the above.
> > >
> > > (i) First, it would be possible to modify your proof to get to a correct proof, but the ways we see to do it would involve incorporating the remaining ideas from the proof in our paper. In particular, defining criticality has to be done carefully, as the problem with your incorrect argument introducing simple-criticality makes clear. Moreover, and crucially, even with our notion of criticality, the true language $L_z$ does not necessarily become critical as soon as $t \geq z$ (as you were attempting to achieve with simple-criticality). Rather, we may have to wait until a potentially later step in the enumeration; our paper accomplishes this in (4.3) via the analysis of the step $t^+$. If you make all these changes, then you would fix the problems with your current proposed proof, but you would also be gradually arriving at all the steps in our current proof.
> > >
> > > (ii) You argue that our explanations have unnecessary length. Given that full proof in our paper is only a few pages, we do not think it is particularly long in an absolute sense, even with complete explanations included. Moreover, given that your reviews have now contained two incorrect attempts at a proof, we would suggest that this indicates how getting the details of the proof right is fairly subtle, and it is easy to inadvertently set things up in a way that leads to errors. That is exactly the kind of situation that we typically think of as calling for complete arguments and explanations rather than abbreviated ones. For example, your later suggested description that "The inclusion queries in Item 1 can be replaced by membership tests (as done by the authors) simply by considering slices of the languages up to $L_t$ containing only strings up to a certain length" is indeed correct at a high level, but it is essentially equivalent to Section 5 of our paper, just with all the details suppressed. Given the subtlety of these arguments, and the ease with which errors can arise, we think it is important for these details to be present; and if you were to fill in these details, you would get back to something essentially equivalent to Section 5.
> > >
> > > On your remaining points, we believe that the feasibility of language generation is a question of fundamental interest, and given that NeurIPS has tracks for theoretical work on the inherent limits to learning, we also believe it is clearly in scope for the conference. The paper discusses on pages 2-3 and again on page 9 some of the potential connections to current issues in language modeling; we agree there are many open questions that can be considered here, and we find the presence of these open questions a benefit of the current direction.
> > >
> > > On the point about the languages $L_i$ being infinite, as noted earlier, we agree that the question becomes technically more complicated when some languages can be finite, and we can indeed discuss this point in a revision. As we also discussed earlier, we think that these added complications arising from finiteness detract from the underlying motivation rather than adding to it. In particular, we'd reiterate the point from our earlier response that the challenge in real language generation problems is not the concern that the training data might have exhausted all possible valid utterances; it is generally understood, both intuitively and on more technical grounds, that there will always be further valid utterances that have not yet been seen. This is exactly the reason to assume that the candidate languages $L_i$ are infinite.

---

> > > > ### Comment · Reviewer_v4Eq · 2024-08-13
> > > >
> > > > Thanks for the explanation of the bug in my second attempt of a simplified proof sketch. Also, thanks for proposing a fix to this approach.
> > > >
> > > > Still, I'm convinced that the exposition of your proof can be significantly improved by taking a more Definition/Statement/Proof approach, and by taking a more algorithmic approach using the terminology of Inclusion Queries and Membership queries. More specifically:
> > > >
> > > > 1) Define a suitable notion of criticality
> > > > 2) State the main theorem in terms of Inclusion Queries and Membership queries
> > > > 3) State the protocol for solving the language generation.
> > > > 4) Prove correctness using the properties of criticality.
> > > > 5) Remove the need for inclusion queries.
> > > >
> > > > The very technical instantiations of each step are not the relevant aspect here. Your own definitions/statements can be rearranged to fit the flow mentioned above. The important thing is that the flow of the proof would be significantly easier to follow in this way, and would save the reader from the need to go back and forth in the argumentation.
> > > >
> > > > That said, even though digging into the proof is something that I find interesting, as explained in my previous reply, the proof exposition is only one aspect of my recommendation. Especially, in my opinion, Reasons 2, 3 and 4 are equally important when considering papers for a conference in the realm of machine learning / neural networks, where pure computability results may not be the best fit.
> > > >
> > > > Unfortunately the replies corresponding to these three other aspects did not make me change my mind. The drawbacks are still there and improving the paper to address these drawbacks would require some work that goes beyond reorganisation and polishing.
> > > >
> > > > So I will maintain my score, despite the interesting discussion in the proof side.

---

> > > > > ### Author Response · Authors · 2024-08-14
> > > > >
> > > > > Thanks for your reply. We understand that we disagree on several of these points, but we appreciate all the thought you put into the proofs and the details of the paper, and your suggestions about different approaches to the main result.

---

### Official Review · Reviewer_4Kah · 2024-07-13

**Soundness:** 4
**Presentation:** 4
**Contribution:** 3
**Rating:** 7
**Confidence:** 4

**Summary:**

This paper revisits a classic topic with a new angle that reveals a more positive outlook on a classically pessimistic result.

Namely, the paper that while identification of a formal language from positive examples is generally not possible, even given countably infinite examples, one *can* always learn a (under) approximation of the language which for non-finite languages is itself infinite!

The paper starts by reviewing why the obvious approach cannot work -- and also reminds the reader why it would violate Gold's classic negative result on language identification.

The paper then constructively proves that language generation is possible in the limit. Further analysis is done for finite collections of formal languages where the learning of the under approximation has an upperbound (specific to the finite collection) independent of the adversaries example strategy.

**Strengths:**

The paper does an excellent job of treating a classic technical subject while still being approachable. In particular, the treatment of the review of the negative results is very well handled and serves a great pedagogical launching point for the constructive proof.

The language generation problem itself is also interesting and the connection to the recent craze in generative AI makes it timely and appropriate for the Neurips audience. While modern LLMs likely exploit more structure afforded by distributional properties, the fact that a stochastic parrot need not in general is interesting.

My reading of the proofs revealed no errors.

**Weaknesses:**

While I overall find the paper to be well paced and easy to read, the final analysis of the algorithm and section 6 felt very curt. This is something that I'm sure could be addressed with an additional page in the camera ready.

**Questions:**

* Are there any implications on this result for common families of languages such as regular languages which have a natural notion of complexity? e.g., representation size. It seems like one could bound how many examples it would take by additionally ordering candidates by complexity.

**Limitations:**

Handled well in the concluding remarks of the paper.

---

> ### Author Rebuttal · Authors · 2024-08-06
>
> Thank you for the review of the paper; we appreciate the comments about the work and the interesting questions.
>
> We agree that with the length constraints of the submission, Section 6 is written in a very compressed format. Indeed, we would plan to provide a more extended discussion of the result and the proof in Section 6 in a version with a higher page limit.
>
> We also agree that there are a number of interesting open directions for considering quantitative versions of the results here for language families with specific structure, like regular languages and context-free languages. For example, when the result of Section 6 is specialized to a set of regular languages or context-free languages, it raises an interesting set of language-theoretic questions about how large the bound $t({\cal C})$ on the required number of examples needs to be as a function of an upper bound $k$ on the number of states in the finite automata that generate the languages in the collection $\cal C$ (with similar questions for corresponding measures of complexity for context-free languages).
>
> Similarly, in the case of the result for infinite collections of languages in Sections 4 and 5, when specialized to regular or context-free languages we agree that your suggestion of ordering languages by increasing complexity may make possible quantitative bounds on the number of examples needed as a function of the position of the true language $K$ in the list.
>
> We would plan to mention these interesting open directions for regular and context-free languages in a revised version of the paper.

---

> ### Comment · Reviewer_4Kah · 2024-08-13
>
> Writing to acknowledge I have read the rebuttal and other reviews.
>
> I still stand by advocating for acceptance and also respectfully disagree with the correctness of reviewer v4Eq’s proposed protocol. It falls for the (alluring) trap laid out in section 3.

---

> > ### Author Response · Authors · 2024-08-13
> >
> > Thanks very much for your reply; we appreciate that you were able to look through the other reviews and rebuttals, and are glad to see your confirmation of the presence of the error in the solution proposed by Reviewer v4Eq.

---

### Decision · Program_Chairs · 2024-09-25

**Decision:**

Accept (spotlight)

**Comment:**

The paper address the language generation problem within the theoretical framework of PAC learning. It shows that according to their definition, language generation is possible in the limit while language learning (old results by Angluin et. al) is not. The reviewing scene was not ideal. One reviewer strongly argued for acceptance, another gave a borderline rank but provided no information,  the 3rd reviewer
suggests  rejection claiming that the paper has an error. This reviewer reviewed an earlier submission of the paper and claimed his previous complaints were not addressed. There was a discussion between the authors and the reviewer. I read the paper I think the paper is important as it bring the foundational theoretical perspective into current LLM culture.